# The Role of Macrophages in the Development of Acute and Chronic Inflammatory Lung Diseases

**DOI:** 10.3390/cells10040897

**Published:** 2021-04-14

**Authors:** Jae-Won Lee, Wanjoo Chun, Hee Jae Lee, Jae-Hong Min, Seong-Man Kim, Ji-Yun Seo, Kyung-Seop Ahn, Sei-Ryang Oh

**Affiliations:** 1Natural Medicine Research Center, Korea Research Institute of Bioscience and Biotechnology (KRIBB), Chungbuk, Cheongju 28116, Korea; 01020835969@kribb.re.kr (J.-H.M.); ksm2906@kribb.re.kr (S.-M.K.); yun@kribb.re.kr (J.-Y.S.); 2Department of Pharmacology, College of Medicine, Kangwon National University, Chuncheon 24341, Korea; wchun@kangwon.ac.kr (W.C.); heejaelee@kangwong.ac.kr (H.J.L.); 3College of Pharmacy, Chungbuk National University, Cheongju 28160, Korea; 4College of Pharmacy, Chungnam National University, Daejeon 34134, Korea

**Keywords:** macrophages, M1/M2 polarization, cytokines, acute/chronic inflammatory diseases

## Abstract

Macrophages play an important role in the innate and adaptive immune responses of organ systems, including the lungs, to particles and pathogens. Cumulative results show that macrophages contribute to the development and progression of acute or chronic inflammatory responses through the secretion of inflammatory cytokines/chemokines and the activation of transcription factors in the pathogenesis of inflammatory lung diseases, such as acute lung injury (ALI), acute respiratory distress syndrome (ARDS), ARDS related to COVID-19 (coronavirus disease 2019, caused by severe acute respiratory syndrome coronavirus 2 (SARS-CoV-2)), allergic asthma, chronic obstructive pulmonary disease (COPD), and idiopathic pulmonary fibrosis (IPF). This review summarizes the functions of macrophages and their associated underlying mechanisms in the development of ALI, ARDS, COVID-19-related ARDS, allergic asthma, COPD, and IPF and briefly introduces the acute and chronic experimental animal models. Thus, this review suggests an effective therapeutic approach that focuses on the regulation of macrophage function in the context of inflammatory lung diseases.

## 1. Introduction

Macrophages are innate immune cells that perform phagocytosis and eliminate pathogens as part of physiological processes [1]. CD64, a high-affinity Fc-γ receptor, is known as a marker of M1 macrophages, and CD163 and CD206 have been identified as major markers of M2 macrophages [2,3]. The expression of these surface makers is closely associated with pathogen phagocytosis and inflammatory responses. Macrophages are abundantly present in the lung microenvironment, where they are mainly found as alveolar macrophages (AMs) and interstitial macrophages (IMs) (Figure 1) [4]. Circulating monocytes are known to act as a new source of macrophages when AMs are damaged [5,6]. AMs are polarized into M1 and M2 phenotype macrophages, whereby M1-type macrophages play a pivotal role in the pro-inflammatory reactions of host defense, and M2-type macrophages contribute to anti-inflammatory responses and tissue remodeling [7] (Table 1 and Table 2). Thus, regulation of macrophages is important for both pro-inflammatory and anti-inflammatory effects. Accumulating evidence has shown that the polarization of macrophages, which is regulated by cytokines, chemokines, and transcription factors, is closely related to the initiation and development of pulmonary inflammatory diseases, such as acute lung injury (ALI), acute respiratory distress syndrome (ARDS), COVID-19 (coronavirus disease 2019)-related ARDS, allergic asthma, chronic obstructive pulmonary disease (COPD), and idiopathic pulmonary fibrosis (IPF) [4,7,8,9,10]. Therefore, the regulation of macrophage polarization and these cell-associated inflammatory molecules may represent a valuable therapeutic approach for treating acute and chronic inflammatory lung diseases. In this review, we will discuss the importance of M1/M2 macrophage roles in the development of ALI, ARDS, allergic asthma, COPD, and pulmonary fibrosis.

## 2. Macrophages in ALI/ARDS

Acute lung injury (ALI) and acute respiratory distress syndrome (ARDS, known as ALI clinical symptoms) are recognized as serious public health issues as they result in respiratory failure and high mortality rates [11]. ALI/ARDS has been known to originate from various factors, such as bacteria, chemicals, and viruses, and are characterized by leukocyte aggregates and alveolar epithelial damages [12,13]. Various pharmacologic treatments, including glucocorticoids, anti-inflammatory agents, and antioxidants, have been tested in clinical trials for ARDS [14]. However, there is currently no effective drug therapy for these disorders.

Neutrophils and macrophages exert host defense against microbial invasion [8,11]. A neutrophil influx into the lungs is a hallmark of ALI/ARDS [15], and activated neutrophils cause inflammatory responses and tissue damage by producing toxic molecules and cytokines [16]. In the pathogenesis of ALI/ARDS, alveolar macrophages (AMs) are polarized into M1/M2 macrophages and participate in the exudative phase, rehabilitation phase, and, finally, the fibrotic phase (Figure 2) [8].

Microbial components are recognized by resident AMs through their pattern-recognition receptors (PRRs), which, in turn, induce nuclear factor-κB (NF-κB)-dependent polarization into M1-type macrophages, leading to initiation of the exudative phase [17]. In this phase, M1 macrophages have been reported to produce inflammatory cytokines, chemokines, and toxic molecules such as tumor necrosis factor-α (TNF-α), interleukin-1 β (IL-1β), IL-6, monocyte chemoattractant protein-1 (MCP-1), macrophage inflammatory protein 2 (MIP-2), inducible nitric oxide synthase (iNOS), cyclooxygenase-2 (COX-2), and reactive oxygen species (ROS), which are associated with the recruitment of inflammatory cells, including monocytes and neutrophils, and the promotion of airway inflammation, antimicrobial activity, and lung tissue injury [8,18]. Researchers report that suppressor of cytokine signaling 3 (SOCS3) is associated with the suppression of macrophage polarization into the M1 phenotype, downregulating the activation of signal transducers and activators of transcription (STAT)1/3, the nuclear translocation of inborn errors of immunity in interferon (IFN) regulatory factor (IRF) 5 and the secretion of pro-inflammatory molecules [19,20,21,22]. Chen et al. reported that c-Jun N-terminal kinase (JNK) is required for M1-type macrophage polarization and macrophage recruitment [8,23,24,25]. Another study emphasized the essential role of JNK and p38 mitogen-activated protein kinase (MAPK) signaling in the development of endotoxin-induced ARDS [26,27]. By contrast, a previous study indicated that the upregulation of JNK expression and c-Myc transcription could promote the M2 phenotype in IL-4-stimulated macrophages [28,29]. These results indicate that NF-κB, SOCS3, STAT1/3, and IRF5 play an important role in the polarization of M1 macrophages and that JNK plays a dual role in M1/M2 macrophage polarization.

After pathogenic or viral virulence factors are removed, M1 phenotype macrophages change into M2 phenotype macrophages during the rehabilitation phase of ALI/ARDS [30]. In this phase, M2 phenotype macrophages are classified into four distinct subtypes (M2a, M2b, M2c, and M2d). The M2a subtype is induced by IL-4 and IL-13, LPS promotes the M2b subtype, IL-10, and transforming growth factor-β (TGF-β) activates the M2c subtype, and the M2d subtype is triggered by tumor-associated factors [31]. It has been reported that STAT6 and IRF4 are involved in M2-type macrophage polarization [15,32]. M2 phenotype macrophages release anti-inflammatory molecules, including IL-10 and TGF-β, in response to Th2-type cytokines (IL-4 and IL-13) and inhibit pro-inflammatory mediators. Therefore, they exert a pivotal role in repairing lung damage [33,34]. Under these conditions, neutrophils are eliminated, regardless of changes in the balance of pro- and anti-inflammatory molecules. [35].

Pulmonary fibrosis, the late phase of ALI/ARDS, is induced by fibroblast proliferation and excessive collagen deposition [31]. In this phase, the trend and severity of pulmonary fibrosis are determined by the balance of M1 and M2 macrophages. Therefore, the control of macrophage polarization may be expected to improve the progression of fibrosis in ALI/ARDS. It has been reported that CXCL10 and matrix metalloproteinases (MMPs) generated by M1 macrophages lead to the promotion of matrix degradation and the inhibition of fibrosis [8]. M2 phenotype macrophage-derived TGF-β results in the promotion of myofibroblast proliferation and extracellular matrix (ECM) deposition [14,36]. Furthermore, growth factors, including MIP-2, vascular endothelial growth factor (VEGF), angiopoietin, and platelet-derived growth factor (PDGF), have been shown to promote collagen deposition [14]. Therefore, the increased levels of M2 macrophages and TGF-β are recognized as hallmarks of fibrosis development, and the persistent production of IL-4 and IL-13 is associated with fibronectin and collagen deposition by promoting M2 phenotype macrophages [37,38]. However, recent results have also shown that M2 macrophages could exert an ameliorative effect on fibrosis by generating arginase 1 [10,39,40]. These results indicate that macrophages contribute to the promotion of the healing of fibrosis [41]. It is necessary to clarify the regulatory mechanism of macrophages in fibrosis.

### Experimental Animal Models of ALI/ARDS

Regarding ALI/ARDS development, endotoxin lipopolysaccharide (LPS) has mostly been used in experimental animal models of ALI/ARDS [42,43,44,45,46,47]. In these animal models, mainly C57BL/6 and BALB/c mice are used, and LPS (0.5–5 mg/kg) is used to induce the development of ALI or ARDS. Similar to observations in preclinical studies of ALI/ARDS patients, where neutrophil/macrophage levels were increased, levels of these cell-derived molecules were also elevated. Therein, the anti-inflammatory agents and antioxidant exerted regulatory effects on neutrophil/macrophage recruitment; the expression of iNOS, TNF-α, IL-6, MCP-1, IL-1β, and TGF-β; and the activation of SOSC3, STAT3, MAPK, and NF-κB. Recent studies in ALI/ARDS animal models have indicated that nuclear factor erythroid-2-related factor 2 (Nrf2) activation, and heme oxygenase-1 (HO-1) induction could ameliorate airway inflammation and macrophage activation [48,49]. A previous report has also shown the ameliorative effect of AM reduction in lung injury [50].

Bleomycin is known to promote TGF-β, fibroblast proliferation, and collagen synthesis, and, therefore, it is widely used for pulmonary fibrosis in preclinical studies of ALI/ARDS [51,52,53,54]. In these animal models, increased levels of fibrotic markers such as α-smooth muscle actin (α-SMA), collagen accumulation, and the generation of M2 polarized macrophages were confirmed. In addition, Li et al. reported that the increase in neutrophil/macrophage numbers and of IL-6 and TNF-α in BALF and TGF-β in lung tissue was confirmed in bleomycin-induced mice, and this increase could be suppressed with antioxidants by upregulating HO-1 induction and Nrf2 activation [55].

## 3. Macrophages in COVID-19-Related ARDS

The continuous spread of coronavirus disease (COVID-19) in 2020 has killed millions of people worldwide [56,57]. Macrophage Toll-like receptors (TLRs) recognize single-stranded RNA (ssRNA) fragments of SARS-CoV-2 [10,57], and this recognition could increase the expression of inflammatory cytokines, chemokines, and growth factors, including TNF-α, IL-1β, IL-6,7,8,9,10, MCP-1, and granulocyte–macrophage colony-stimulating factor (GMCSF) in cases of COVID-19 [58]. This elevated expression of inflammatory molecules is related to activation of the NF-κB pathway [59,60,61], indicating that the NF-κB pathway may be a valuable therapeutic target in COVID-19.

Recent studies have indicated that increased levels of macrophage-associated cytokines lead to lung inflammation and cytokine storm, and these increases in cytokines are closely correlated with increased disease severity [62,63,64,65]. The high increase in pro-inflammatory cytokines, including TNF-α and IL-1β, and in chemokines, such as IL-8, have been confirmed in the upper respiratory tract [62,66]. Furthermore, it has been reported that macrophages exert a strong inflammatory reaction in the lower airways compared to the upper respiratory tract [62]. These results suggest that macrophages-derived pro-inflammatory cytokines play an essential role in cytokine storms, and thus these molecules are thought to be associated with COVID-19-associated ARDS mortality [66]. To ameliorate the complications induced by cytokine storms, targeting IL-6 studies are currently being conducted [67,68,69]. Researchers have reported that the MAPK pathway is well recognized in viral infection [70]. Another study has also focused on MAPK activation in COVID-19 [71].

Pulmonary fibrosis, along with cytokine storms, accompany SARS-CoV-2 [67]. M2-type macrophages are thought to play an important role in the fibrosis of COVID-19. The therapeutic potential of histone deacetylase (HDAC) inhibitors, such as suberanilohydroxamic acid (SAHA) in fibrosis, was reported in a previous review paper [72,73,74]. HDAC6 inhibition inhibits M2-type macrophages and type I collagen protein expression induced by TGF-β in lung fibroblasts [75,76]. Thus, these reports indicate that HDAC inhibitors could ameliorate fibrosis induced by M2 macrophages in COVID-19. However, the actual role of M2 macrophages in fibrosis is not well established. Thus, it is necessary to clearly investigate the role of M2 macrophages in this response.

### Experimental Animal Models of COVID-19

Different animal models have been used to characterize viral pathogenesis and for the evaluation of vaccines and antiviral drugs [77]. K18-human angiotensin-converting enzyme 2 (hACE2) transgenic mice were used to study COVID-19, and in this model, severe neuronal and lung damage was confirmed in response to SARS-CoV-2 infection [78]. Thus, this model is considered to be among the best available experimental mouse models for COVID-19. Golden Syrian hamsters have been recommended as a COVID-19 animal model in which viral replication and weight loss were observed following infection with SARS-CoV-2 virus [79]. Severe lymphoplasmacytic perivasculitis has been observed in ferret lungs after infection [80]. Munster et al. reported that COVID-19 symptoms, such as viral replication, were found in rhesus monkeys infected with the live virus [81].

## 4. Macrophages in Allergic Asthma

The worldwide prevalence of allergic asthma, a chronic inflammatory disorder, is still high; specifically, its incidence has steadily increased in developed countries [82,83]. Airway inflammation, mucus hypersecretion, and airway hyperresponsiveness (AHR) are the major symptoms of allergic asthma and are profoundly impacted by Th2 cytokines, such as IL-4, IL-5, and IL-13, and chemokines [4,84].

The increased infiltration of eosinophils in the lungs is recognized as a hallmark of allergic asthma, resulting in excessive mucus secretion and AHR [46]. C–C motif cytokine ligand (CCL)11/eotaxin, specifically eosinophil chemokine, is crucial for eosinophil recruitment into inflammatory sites in allergic asthma [85,86]. CCL11 is thought to be produced by Th2 cells via IL-4/IL-13 signaling [87], and the suppression of IL-4 and IL-13 results in decreased not only CCL11 expression but also reduced pulmonary eosinophilia [88,89].

Th2 cytokines, such as IL-4 and IL-13, are major inducers of M2 macrophage polarization, and M2-type macrophages produce inflammatory cytokines and eosinophil-recruiting chemokines affecting airway inflammation, remodeling, and mucus hypersecretion and resulting in decreased lung function (Figure 3) [4,7,90]. iNOS-induced NO promotes mucus secretion in allergic asthma [91]. Mucin 5AC (MUC5AC), an integral component of mucus in the airway, is significantly upregulated in the pathogenesis of allergic asthma, and its increase is related to airway obstruction [92]. Chung et al. reported that the activation of cAMP response element-binding protein (CREB) leads to MUC5AC expression and mucus secretion [93]. Accumulating evidence has reported that Th2 cell-induced immunoglobulin (Ig) binds to IgE receptors on the surface of mast cells which, in turn, produce histamine and leukotrienes [4,90]. Recently, IL-33 has been reported to exert a biological role in the pathogenesis of human diseases; specifically, its critical role in allergic asthma has recently been confirmed [94,95]. In this study, airway epithelial cell-derived IL-33 following exposure to an allergen was shown to induce airway inflammatory response with increased cytokines/chemokines and M2 macrophage polarization. This study also reported that IL-33 induces mast cell activation and histamine secretion in the pathogenesis of allergic rhinitis. Another study also reported that TNF-α affects NO generation, which could induce inflammatory cell recruitment and Th2 immune response (IL-4 and IL-5), resulting in eosinophil recruitment and IgE production [96]. IL-4 and IL-13 cause M2 macrophages to express TGF-β, which, in turn, induces the proliferation of airway smooth muscle cells, fibrosis, and airway remodeling in allergic asthma [97,98,99,100,101]. Thus, the increased level of TGF-β1 is closely associated with airway narrowing and airflow limitation and reflects severe persistent asthma in allergic asthma [97,98,99,100]. miRNAs have been proposed as potential targets for pulmonary fibrosis [102]. Collectively, the increase in M2 macrophage polarization in the lungs reflects asthma severity, leading to the development of an allergic inflammatory response. Therefore, modulating the polarization of macrophages may be beneficial in the treatment and amelioration of allergic asthma.

Accumulating reports have shown that IL-4 and IL-13 lead to activation of the JAK–STAT6 signaling pathway, inducing the M2 phenotype in macrophages [90,103,104]. Increased levels of NO and iNOS are induced by the NF-κB signaling pathway in allergic inflammation [90,105,106]. Recently, Wang et al. reported that IL-37 administration exerts an anti-inflammatory effect on allergic inflammation by downregulating eosinophil recruitment; IL-4, IL-5, IL-6, IL-17a, and IL-10 production; and the mRNA expression of GATA3, RORγT, and Foxp3 [107]. Thus, this study suggested that IL-37 may suppress Th2 response via the inhibition of STAT3 and STAT6 activation. These results indicate that the modulation of JAK–STAT6 and NF-κB activation could ameliorate allergic airway inflammation and mucus secretion by regulating M1/M2 macrophages and suppressing inflammatory molecules. Furthermore, the suppression of IL-33 and induction of IL-37 could exert anti-inflammatory activity in allergic asthma.

### Experimental Animal Models of Allergic Asthma

BALB/c mice have mainly been selected in experimental animal models of allergic asthma [46,91,108,109,110]. Ovalbumin (OVA) has been used as allergens in these models, which results in a significant increase in eosinophil/macrophage numbers, Th2 cytokines (IL-4, IL-5, and IL-13), and TNF-α in BALF, and IgE and histamine in serum [46,109,110,111]. OVA has also been shown to induce the upregulation of mucus secretion in lung tissues in these models. The increased levels of enhanced pause (Penh) have been confirmed by OVA exposure in these models. These responses have been accompanied by the activation of JAK1–STAT6, MAPK, and NF-κB signaling pathways. House dust mites (HDMs) have been reported as allergens that induce Th2-type inflammation in allergic asthma [112]. In these studies, the airway inflammatory response has been accompanied by the activation of MAPK and NF-κB signaling pathways [113]. Interestingly, the suppression of these signaling pathways has led to amelioration of the symptoms of allergic asthma via inhibition of inflammatory cell recruitment, cytokine production, and mucus hypersecretion in both OVA- and HDMs-induced experimental animal models. Recently, Lei et al. showed that IL-37 administration exerts an ameliorative effect on OVA-induced allergic asthma mice by suppressing the levels of eosinophil numbers and the generation of IL-4, IL-13, IL-17, CCL11, and histamine [114].

## 5. Macrophages in COPD

COPD prevalence and mortality have largely increased worldwide [115]. COPD is characterized by chronic bronchitis, oxidative stress, mucus hypersecretion, emphysema, and persistent airflow limitation [41,116,117,118]. Cigarette smoking is known to be the greatest risk factor for COPD [113,114,115]. Guiedem et al. reported that respiratory infection makes breathing much more difficult and leads to further lung damage in the pathogenesis of COPD [118].

Increased levels of neutrophils have been identified in the bronchial wall, lumen, and sputum of COPD patients [119,120,121]. Neutrophil elastase (NE) is highly prevalent in the airways of COPD patients and is known to induce MUC5AC mucin, which is the primary component of mucus in the airway [122,123]. It has also been reported that the activation of TGF-α-dependent epidermal growth factor receptor (EGFR) and the production of ROS are related to NE-induced MUC5AC in bronchial epithelial cells [124,125]. Wang et al. reported that mucin gene MUC5AC expression is regulated by the activation of CREB [126]. Neutrophil-derived elastase causes the decomposition of elastin, resulting in emphysema [127]. Therefore, the release of elastase reflects the important role of neutrophils in the emphysema of COPD. Antunes et al. also reported that increased levels of NE and ROS promoted airway inflammation, emphysema, and lung function damage [127]. A previous study also indicated that macrophage-induced eosinophil efferocytosis is closely related to the severity of COPD [128]. Increased levels of MCP-1 have been confirmed in sputum samples of COPD patients [129]. Macrophage has been shown to produce MCP-1, which, in turn, affects neutrophil recruitment [118,130]. These results indicated that macrophages play an important role in the development of COPD by inducing neutrophil influx.

Cumulative evidence indicates that increased numbers of macrophages are found in the sputum and lungs of COPD patients, suggesting this is related to the severity of the disease [131,132,133,134]. AMs have been found in damaged regions of the lungs and play an important role in the development of chronic bronchitis and emphysema by regulating monocyte and neutrophil recruitment in the pathogenesis of COPD (Figure 4) [135,136]. M1 and M2 phenotype macrophages have been found in the lungs of COPD patients [137].

LPS, a contaminant of cigarette smoke, may induce M1 macrophage polarization [138,139]. This is supported by a previous study indicating that elevated levels of inducible nitric oxide synthase (iNOS) expression were found in the AMs of COPD patients [140,141,142]. iNOS-induced NO and ROS are able to induce oxidative stress [143]. Thus, M1 polarization-mediated iNOS promotes oxidative stress in COPD. Furthermore, accumulating evidence from research shows that the levels of IL-1β, IL-6, IL-8, and TNF-α are upregulated in COPD [140,144,145,146,147]. These results are a reflection of M1-type macrophage-derived cytokines being related to the development of COPD. These results indicate that M1 macrophage-induced cytokines contribute to the pathogenesis of COPD.

A recent study showed that M2 phenotype macrophages are dominant in the BALF of COPD patients with increased cytokines such as IL-4, IL-13, IL-8, and IL10 [148,149]. Several groups reported that smoking promotes M2 polarization of AMs and the expression of MMP12 [70,71,138]. IL-4-induced M2 macrophages produce MMP12, which has been found to play a pivotal role in emphysema [138,150,151]. A significant increase in IL-13-mediated M2 polarization and MUC5AC has been shown in the lungs of patients with severe COPD [152]. M2 macrophage-induced IL-8 causes the production of neutrophil-elastase, which, in turn, promotes MUC5AC expression, leading to mucus hypersecretion and airway obstruction [91,149]. It has also been shown that IL-33 levels were elevated in the nasal cavities or epithelium of COPD patients [95,152].

As was reported for ALI, ARDS, and allergic asthma, airway fibrosis was also found to be associated with TGF-β pathways in COPD [9]. In this report, the involvement of the M2 type-dependent TGF-β/small mothers against the decapentaplegic (Smad) pathway was confirmed in fibrotic remodeling. However, it is necessary to clarify the functional role of M2 macrophages in the development and progression of fibrosis.

The expression of pro-inflammatory cytokine was induced via the p38 MAPK pathway in response to cigarette smoke [153,154], and increased p38 activity was confirmed in alveolar macrophages derived from COPD patients [155]. In these studies, the administration of corticosteroid with p38 MAPK inhibitor promotes synergistic anti-inflammatory effects on endotoxin-stimulated cytokines by AMs from COPD patients. As in asthma, higher levels of activated NF-κB are observed in the bronchial biopsies and inflammatory cells of COPD individuals [122]. IκB phosphorylation and ubiquitination lead to NF-κB activation and nucleus translocation, inducing inflammatory cytokines such as IL-1β, IL-6, and TNF-α in COPD [122]. Compared with healthy non-smoking individuals, IκBα levels in the lung tissue of smokers and COPD patients are also lower [156]. Therefore, the inhibition of p38 MAPK and NF-κB activation causes the suppression of inflammatory molecules, and, therefore, p38 MAPK and NF-κB signaling represent potential effective targets in COPD therapy.

### Experimental Animal Models of COPD

An experimental animal model of COPD has been built using cigarette smoke (CS), cigarette smoke extract (CSE), and CS with elastase or CS with LPS [157,158,159,160]. In these experimental settings, the remarkable expression of TNF-α, IL-6, IL-1β, MCP-1, ROS, and elastase and the significant increase in neutrophil and macrophage numbers were confirmed along with the progression of airway inflammation, mucus secretion, and emphysema. The outstanding existence of inflammatory cells in the nearby airway was characterized in this animal model. This change was accompanied by NF-κB and MAPK activation.

Kubo et al. also reported that exercise could ameliorate the symptoms of COPD by reducing emphysema in an experimental animal model of COPD [157]. In this study, the enhanced upregulation of Nrf2 and HO-1 expression was identified in the lungs of the exercising and smoking groups compared to the control or smoking group, and this model may support the importance of exercise for a better prognosis in COPD patients. Recently, Lerner et al. indicated that the ablation of CXCR2, an IL-8 receptor, ameliorates airway inflammation and DNA damage by suppressing neutrophil recruitment and levels of inflammatory molecules such as cytokines and chemokines in CS-exposed mice [161]. He et al. reported the characterization of M2 macrophages in the lungs of COPD-exposed mice [9]. In their study, IL-4-stimulated RAW264.7 macrophages were polarized into M2 macrophages with activation of the TGF-β/Smad pathway, indicating that CS may promote the development of M2-type macrophage and activation of the TGF-β/Smad pathway.

## 6. Macrophages in IPF

IPF is a chronic lung disease that is characterized by progressive pulmonary scarring, fibrosis, and shortness of breath [162]. It has been reported that the incidence of IPF has globally increased [163]. IPF mortality is high, and its median survival time is only two to three years after diagnosis [163,164]. Currently, US Food and Drug Administration (FDA)-approved drugs, such as pirfenidone and nintedanib are available for IPF treatment [165,166,167]. Pirfenidone is a small molecule that inhibits the expression of inflammatory cytokines, such as TNF-α, and downregulates TGF-β1-mediated collagen production and fibroblasts proliferation [168]. Nintedanib, a small-molecule tyrosine kinase inhibitor, has been proven targeting, fibroblast growth factor (FGF), PDGF, and VEGF [167,169]. However, new therapeutic strategies that can better improve pulmonary fibrosis and provide survival benefits are still needed.

Misharin et al. reported that profibrotic genes (Arg1 and MMP13) are increased in monocyte-derived AM in the progression of lung fibrosis [170]. A previous study reported that the Wnt signaling genes are overexpressed in the lungs of patients with IPF [171]. The activation of TLR2 was shown in IPF patients [172,173]. Cumulative evidence has shown that the M2 phenotype rather than the M1 phenotype is dominantly discovered in the lungs during IPF progression [173,174,175,176,177]. It is generally known that M2 macrophages promote lung fibrosis progression [174,175,176]. M2 macrophage-derived TGF-β leads to promote lung fibrosis, and M2 macrophage depletion leads to amelioration of fibrosis [178]. M2 macrophage polarization is induced IL-4, IL-13 and IL-33 [37,38,94,95]. The administration of anti-IL-33 antibody or the deletion of the interleukin-33 receptor (ST2) ameliorates pulmonary fibrosis in IPF-like mouse [178]. Serum amyloid P (SAP) plays an important role in the elimination of apoptotic and necrotic debris and the suppression of lung fibrocyte accumulation and collagen deposition [179]. A significant reduction in SAP levels was confirmed in IPF patients, and SAP administration exerts antifibrotic effects by suppressing M2 macrophages [179]. Collectively, targeting M2 macrophages could be a feasible therapeutic strategy in clinical treatment for IPF patients. However, the exact mechanisms for balancing the M1/M2 phenotype in IPF etiology have not yet been addressed. Thus, further studies that elucidate the function of M2 macrophages and their mechanisms of action are needed for effective therapeutics of IPF.

### Experimental Animal Models of IPF

Bleomycin has mostly used for pulmonary fibrosis in an experimental animal model of IPF [175,180,181,182,183,184]. In these studies, the increased levels of M2 macrophages, T-cell immunoglobulin and mucin domain 3 (TIM-3), CD163, TGF-β1, TNF-α, IL-1ra, IL-10, MCP-1, Arg1, α-SMA, fibronectin, collagen, and TLR2 were detected. Murray et al. reported that SAP treatment attenuates pulmonary fibrosis by suppressing M2 macrophage phenotype and collagen deposition in an animal model of bleomycin-induced pulmonary fibrosis [184]. Li et al. showed that the inhibition of TLR2 attenuates bleomycin-induced pulmonary fibrosis [181]. Recently, it has been reported that antioxidants could ameliorate bleomycin-induced pulmonary fibrosis by upregulating Nrf2 activation and HO-1 expression in animal models of IPF [185,186].

## 7. Conclusions

In this review, we summarize macrophage-induced inflammatory responses in inflammatory lung diseases. Collectively, changes in the phenotype of macrophages and their derived molecules are closely associated with the initiation and acceleration of various acute or chronic pulmonary diseases, including ALI, ARDS, COVID-19, allergic asthma, COPD, and IPF. We hope that understanding the function of macrophages and their involvement in the mechanisms underlying these diseases could allow the function of these cells to be regulated as part of a potential strategy to inhibit the progression of inflammatory disease. However, further investigation is needed to clarify the role of macrophage polarization and the associated response in these diseases.

## Figures and Tables

**Figure 1 cells-10-00897-f001:**
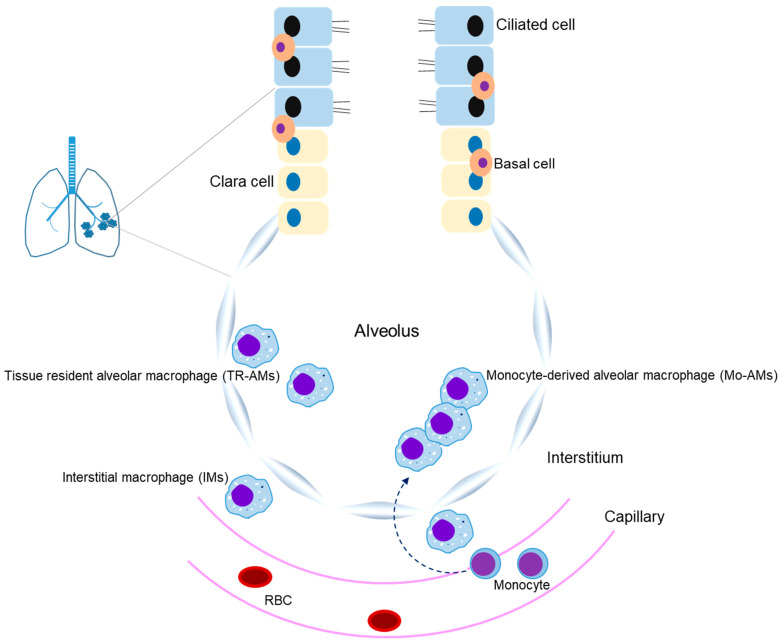
The macrophage population in the lung. Alveolar macrophages (AMs) and interstitial macrophages (IMs) reside in the lungs. AMs are in close contact with the epithelial cells of alveoli, and IMs reside in the parenchyma between the microvascular endothelium and alveolar epithelium. When AMs are damaged, circulating monocytes in the capillaries are then recruited to the lungs and transformed into AM-like cells.

**Figure 2 cells-10-00897-f002:**
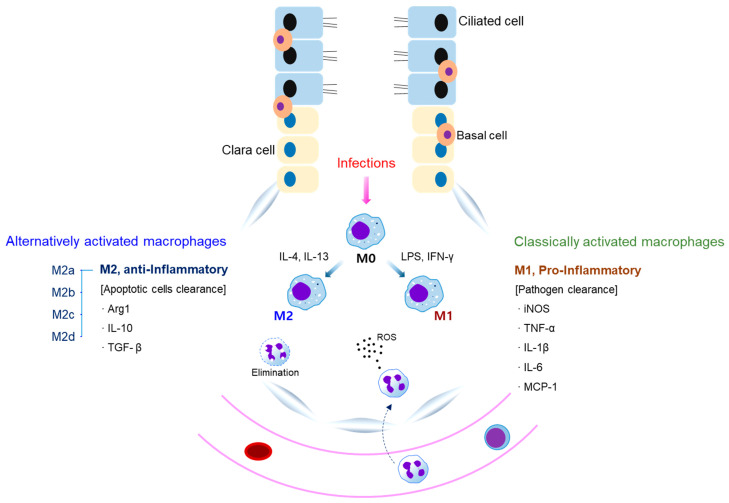
The role of macrophages in ALI/ARDS. Bacterial infection induces the development of ALI and ARDS. Under these conditions, AMs can be classified as M1 and M2 macrophages. Th1 cytokines, such as IFN-γ, and lipopolysaccharide (LPS), induce M1 phenotype macrophages, which produce iNOS, TNF-α, IL-6, and MCP-1, which are responsible for the pro-inflammatory, chemotaxis, radical formation, matrix degradation, and antimicrobial activities during the pathogenesis of ALI/ARDS. Th2 cytokines, such as IL-4 and IL-13, induce M2 phenotype macrophages (divided into M2a, M2b, M2c, and M2d), and these cells produce anti-inflammatory molecules, such as IL-10 and TGF-β, in ALI/ARDS.

**Figure 3 cells-10-00897-f003:**
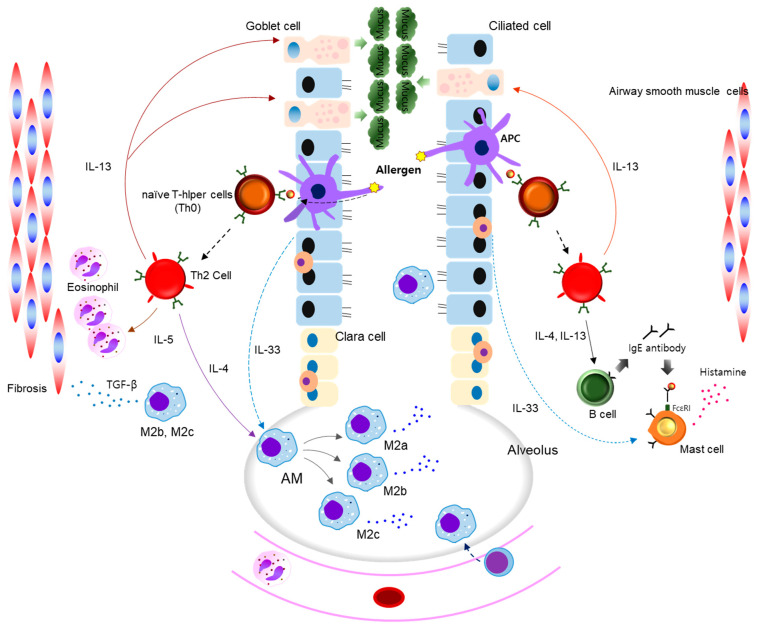
The role of macrophages in allergic asthma. Antigen-presenting cells (APCs) recognize allergens and initiate the allergic cascade. Under these conditions, naïve CD4+ T cells differentiate into Th2 cells. Th2 cell-mediated IL-4 and IL-13 generation lead to the induction of M2 phenotype macrophages. M2 macrophages are classified into subdivisions (M2a, M2b, and M2c). Macrophage-derived inflammatory cytokines and chemokines promote eosinophil recruitment and allergic airway inflammatory responses. Macrophage-derived TGF-β also promotes the proliferation of airway smooth muscle cells and fibrosis. IL-5 induces eosinophil influx and activation. Eosinophil-derived CCL11 leads to eosinophil recruitment. IL-13 induces goblet cell hyperplasia and mucus hypersecretion. This increased secretion contributes to the airflow obstruction. Airway epithelial cell-derived IL-33 promotes levels of M2 macrophages and stimulates mast cells which, in turn, leads to the generation of histamine and leukotrienes. This series of processes leads to airway inflammation and remodeling in addition to mucus overproduction in allergic asthma.

**Figure 4 cells-10-00897-f004:**
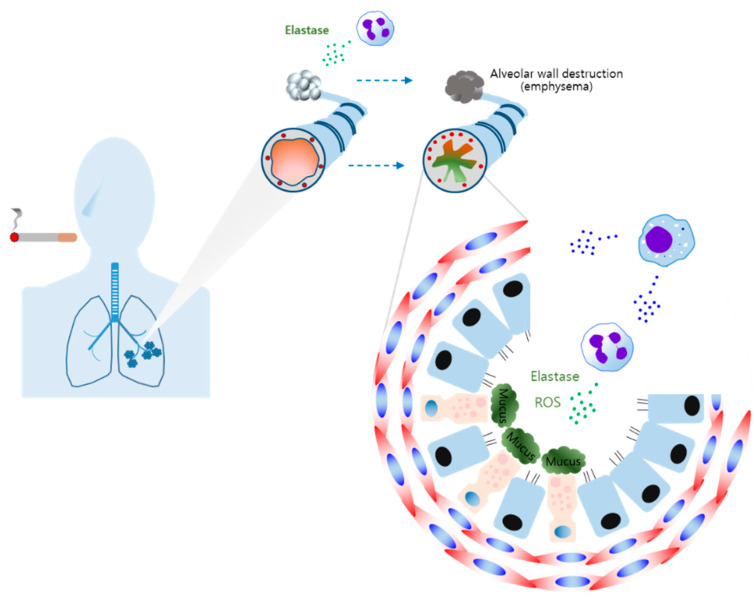
The macrophages in chronic obstructive pulmonary disease (COPD). Tobacco smoking causes airway inflammation, emphysema, and airway remodeling. Neutrophils and macrophages are key inflammatory cells in the pathogenesis of COPD. Neutrophil-derived ROS and elastase promote mucus hypersecretion and emphysema. Macrophages secrete inflammatory cytokines and chemokines, promoting neutrophil recruitment and levels of neutrophil-associated molecules, such as elastase and ROS. Macrophages also are involved in fibrotic remodeling.

**Table 1 cells-10-00897-t001:** The type of macrophages and their role in ALI/ARDS.

Subtypes	Production	Participation
**M1**	iNOS, TNF-α, IL-1β, IL-6, MCP-1	Pro-inflammatory, Neutrophilic inflammation, Tissue injury
**M2**	Arg1, IL-10, TGF-β	Anti-inflammatory, Phagocytosis, Tissue repair and remodeling

**Table 2 cells-10-00897-t002:** The type of macrophages and their role in allergic asthma.

Subtypes	Production	Participation
**M2a**	IL-10, IL-1ra, TGF-β	Allergic inflammation
**M2b**	IL-1, IL-6, IL-10, TNF-α	Tissue remodeling, Fibrosis
**M2c**	IL-10, TGF-β	Anti-inflammatory, Phagocytosis, Tissue remodeling, Fibrosis

## Data Availability

No new data were created or analyzed in this study. Data sharing is not applicable to this article.

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
