# Peer review of "The Role of Macrophages in the Development of Acute and Chronic Inflammatory Lung Diseases"

_cells, 2021, doi:10.3390/cells10040897_

Round 1

Reviewer 1 Report

Lee et al., have reviewed the “The Role of Macrophages in the Development of Acute and Chronic Inflammatory Lung Diseases”. This review constructed firmly and well described in a timely manner.

Although no significant concerns, some queries and suggestion will improve the quality of the manuscript for publication.

Abstract: Throughout the manuscript, animal model are discussed; however, this information is missing in the abstract.

Expand SARS-CoV-2 in the abstract.

Similarly, expand some other acronyms at first instances, MAPK, MMPs, Nrf2, HO-1, HDAC, hACE2, MUC5AC, OVA, NO, Samd, and others.

Page 5, second paragraph: This reference may be useful https://doi.org/10.1016/j.nantod.2020.101051

Page 6, second paragraph: Th12 response --- is this correct?

Page 7: “HDMS-induced” ----- HDMs-induced

Page 7, figure 3 legend second line: “Th2, Th2, and Th17” why two times Th2.

Figure 3: Left-hand side of the figure looks like the allergen directly binds to the naïve T cells, which is not correct.

In the same figure, do authors mean M2 subsets are secreting the following cytokines, M2a – TGF-β; M2b – TNF-α, IL-6; M2c – IL-10, or these cytokines polarising the circulating monocytes into M2 subsets. Please indicate the arrows in the figure and describe the content in the legend.  

Page 8, second paragraph “Alveolar macrophages (AMs)” ---- AMs

Page 8, last paragraph “p38 mitogen-activated protein kinase (p38 MAPK)” ---- p38 MAPK; should have expanded earlier.

By observing figure 2, it is clear that AM secrets both Th1 and Th2 type cytokines. Is that correct? How come both can be secreted simultaneously, or it depends on the type of stimuli. Please provide exact information in the figure and legend.

Throughout the manuscript, some typos need to be corrected, e.g., IL1-β, MCP1, IL-β, and others.

Author Response

We appreciate your professional comment very much.

Reviewer 2 Report

In the current review article, Jae-Won Lee et al. summarizes the role of macrophages in acute and chronic lung disease. Specifically, the authors attempt to describe the collective changes in the phenotype of macrophages and the role of their derived molecules in the initiation and acceleration of various acute or chronic pulmonary diseases, including ALI, ARDS, COVID-19, allergic asthma, and COPD. However, the current manuscript requires a major revision for the following reasons as mentioned in my comments.     

  1. Authors used a broader title to describe the role of macrophages in the development of acute and chronic inflammatory lung diseases. Although, the current manuscript describes their role in ALI/ARDS, COPD, and Asthma. However, the role of alveolar macrophages has been well established in IPF disease initiation and progression. The authors failed to mention the role of alveolar macrophages in IPF in the entire manuscript. It is important to include a section describing the role of macrophages in IPF and its current research status.
  2. In the current manuscript the authors failed to describe the characteristic pathophysiology of ARDS/ALI lung disease.
  3. Figure 1 cartoon is a miss representation of a normal human lung cartoon. The schemata give an impression that all the conducting regions near alveoli only consist of ciliated cells (The authors did not describe what these cells are?) Additionally, the authors stated that AM resides in close contact with epithelial cells of alveoli. However, a better cartoon representing alveolar epithelial cell types and the macrophage position with respect to alveolar epithelial cells would help the reader to better understand the localization of alveolar macrophages in the lungs. Authors can refer to prior publications to draw a better cartoon. The same applies to Figure 2.
  4. Figure 3. cartoon again miss representing the conducting airway. The authors show one side of the airway contains only ciliated cells and the other side consist of a goblet and ciliated cells. Again authors can refer to the existing literature in the Asthma research to draw a better cartoon.
  5. A succinct table highlighting the type of macrophages and their role in each disease condition and cytokine types involved in diseases would make the manuscript message more visible to the larger lung research community.
  6. Disease Short forms are redundantly described in each section after the introduction.
  7. COPD is characterized by chronic bronchitis, oxidative stress, mucus hypersecretion, emphysema, and persistent airflow limitation, and cigarette smoking has been suggested as the major cause of COPD. This sentence is confusing as both the characteristics and cause of COPD are mixed.
  8. Sentence “These results suggest that pro-inflammatory macrophages exert an essential role in cytokine storm and ARDS, which is thought to be the main cause of COVID-19”. This sentence is incomplete. It is not clear whether the authors intended to state the cause of the COVID-19 disease or the cause of COVID-19 deaths?   

9. Sentence “The therapeutic potential of HDAC inhibitors in fibrosis was reported in a previous review paper”. Here, the authors cited a review paper. Instead, authors can describe the original research and cite the related paper.  

Author Response

(The authors gave the same response as above.)

Round 2

Reviewer 2 Report

The authors addressed the comments promptly.